# Does the COVID-19 Pandemic Affect Labor-Related Anxiety and Prevalence of Depressive Symptoms in Pregnant Women?

**DOI:** 10.3390/jcm11216522

**Published:** 2022-11-03

**Authors:** Agnieszka Wikarek, Agnieszka Niemiec, Małgorzata Szymanek, Mateusz Klimek, Justyna Partyka-Lasota, Kamila Dudzik, Tomasz Wikarek, Krzysztof Nowosielski

**Affiliations:** 1Students’ Scientific Society, Department of Gynecology and Obstetrics in Katowice, Medical University of Silesia, 40-752 Katowice, Poland; 2Department of Gynecology and Obstetrics, Medical University of Silesia, 40-752 Katowice, Poland; 3Department of Gynecology, Hospital in Pszczyna, 43-200 Pszczyna, Poland

**Keywords:** COVID-19, SARS-CoV-2, maternal health, newborn health, labor anxiety, depression

## Abstract

The COVID-19 pandemic undoubtedly had significant effects on women’s health and the course of pregnancy. The aim of this single-center study was to explore the impact of the COVID-19 pandemic on adult pregnant and postpartum women’s mental health, as well as to identify factors associated with depressive symptoms, anxiety and fear of delivery. The 465 women included in this questionnaire-based cohort study were divided into two groups: one (controls) of women who gave birth before (*n* = 190), and the second who were pregnant and delivered during the pandemic (*n* = 275). The COVID-19 pandemic affected the severity of self-reported anxiety regarding childbirth (mean scores 2.7 vs. 2.36, *p* = 0.01). The depression (19.84 ± 13.23) and anxiety (16.71 ± 12.53) scores were higher in pregnant women during the COVID 19 pandemic, compared to women who gave birth before the pandemic (8.21 ± 7.38 and 11.67 ± 9.23, respectively). These findings demonstrate the magnitude of the pandemic’s impact on women’s mental health, and actions to improve the mental health of pregnant women in Poland may be crucial for maternal and fetal well-being.

## 1. Introduction

The World Health Organization (WHO) declared COVID-19 a pandemic on 11 March 2020, and within less than two years, 424,822,073 infections worldwide had been confirmed including 5,890,312 deaths (data on 22 February 2022) [1,2,3]. In the case of Poland, the number of infections in the general population was 5,582,217, including 110,157 deaths [4]. In terms of the number of births in Poland, a report by the Central Statistical Office estimated that 147,675 babies were born in 2020 during the second and third waves of the pandemic [5].

Globally, restrictions were put in place in all countries to reduce infections and reduce the spread of SARS-CoV-2. These included travel restrictions, temporary closures of certain facilities, and reduced human contact. Among the groups at risk for severe COVID-19 were pregnant women, who were given special care. In particular, the risk of infection was prevented by limiting hospital visits [6]. 

The impact of the COVID-19 pandemic may be a key element affecting obstetric health, delivery, and the postpartum period. A recently published study showed an association between SARS-CoV-2 infection and increased rates of cesarean sections and postpartum complications [6].

The risk of mental illness is significantly increased during pregnancy [7]. Although the pregnancy itself does not seem to increase that level, other important factors have to be underlined such as younger age of woman, being single, stressful life, high risk pregnancy with complications, and poor overall health status [8]. Similarly, the COVID-19 pandemic cannot be underestimated; before COVID-19, as many as between 13% and 22% of pregnant women experienced anxiety and between 8% and 12% suffered from depression [2]. Furthermore, pregnant women with a medical history of mental disorders may experience higher levels of anxiety [7]. Recent reports suggest that giving birth during a pandemic is related to an increased risk of posttraumatic stress disorder and postpartum depression [9,10]. Moreover, pregnant and postpartum women are at greater risk of more severe COVID-19 [10,11]. In contrast, social support, healthy behavior pattern and positive appreciation of pregnancy are protective factors, especially during the COVID-19 pandemic [10].

The link between the COVID-19 pandemic and anxiety levels in pregnant women is currently being investigated. The results of contemporary studies show that fear of being infected, fear of not being prepared for delivery if COVID-19 infection is detected, social isolation, a lack of physical exercise due to isolation, difficulties in accessing medical care, fear of being infected when having medical routine follow-up visits, and being responsible for infant infection (associated with guilt, and being blamed or stigmatized by society) are believed to be risk factors for anxiety in pregnancy. All these factors are believed to be responsible for increases in the prevalence of perinatal anxiety by up to as much as between 43% and 71% [12,13].

The underestimation of modifiable behavioral and cognitive factors associated with a pandemic can have a significant impact on women’s mental health during a pandemic. Special precautions could be undertaken to minimize all risk factors and improve the health of pregnant women. For that reason, the aim of the study was to determine the prevalence of symptoms of anxiety and depression during pregnancy among those delivering in our department, and to identify the socio-demographic and psychosocial factors that could modify the level of labor-associated anxiety in pregnant women during the third wave of the COVID-19 pandemic.

Based on the result of the previous studies and meta analysis [2,12,13,14,15,16] we hypothesized that: The level of anxiety will be greater in those delivering during the COVID-19 pandemicPresence of depressive symptoms would be a risk factor for a higher level of labor-related anxietyHigher level of trait anxiety would be correlated with higher level of labor-related anxietyLower self-esteem will be associated with higher level of peripartum anxietyHaving epidural analgesia in previous pregnancy will be a protective factor for peripartum anxiety.Partner support will decrease the level of peripartum anxiety.

## 2. Materials and Methods

### 2.1. Study Design and Participants

In total, 1664 women delivering at the Department of Obstetrics and Gynecology of the University Clinical Center of the Medical University of Silesia in Katowice before the COVID-19 pandemic (pre-COVID-19 group) and 1748 delivering during the pandemic (COVID-19 group) were eligible for this questionnaire-based cohort study. The inclusion criteria were physiological pregnancy, no contraindication to vaginal delivery, and willingness to participate in the study. Since high-risk pregnancy as well as complicated course of pregnancy are known risk factors for depressive symptoms and anxiety [12,14], all pregnant women who had shown a risk of miscarriage or preterm delivery, those delivering before 37 + 0 gestation weeks (delivering at term), and those with multiple pregnancy, comorbidities other than thyroid dysfunction, diabetes mellitus, hypertension or pregnancy-induced hypertension, were excluded from the study. Being in a relationship was not an exclusion criterion. 

Out of all the eligible women, 349 met the inclusion criteria before and 424 during the pandemic. However, 83 and 97, respectively, were delivered via unplanned C-section, whereas 76 and 52, respectively, did not return the questionnaires. Finally, the study sample included 465 women. The response rates were 71.4% and 84.1%, respectively. 

The control group consisted of women who gave birth before the COVID-19 pandemic (*n* = 190) between January 2019 and March 2020. The investigated group comprised women who were pregnant and gave birth during the pandemic (*n* = 275) between April 2021 and November 2021.

Based on the results of previous studies, the prevalence of anxiety and depression during the COVID-19 pandemic varies between 43–71% and 25–37%, respectively [12,13]. As the frequency of severe symptoms is low, the minimal expected difference in severe anxiety was used for sample size calculation. In that context, to achieve the minimum required number of participants, a sample size calculation was performed: this showing that to detect a 10% difference in the prevalence of severe anxiety based on the Brief Self-Rating Scale of Depression and Anxiety between cases and controls, 441 total subjects would be required, with the power of 90% and 95% Confidence Level. Similarly, within the same parameter, a minimum sample size of 437 respondents was required to detect a difference of 8% in the prevalence of severe peripartum anxiety based on a Labor Anxiety Questionnaire scores.

A paper and pencil version of the study questionnaire was given to participants twice: up to 7 days before the delivery and one day postpartum in both groups. 

All participants agreed to participate in the study and signed a written consent form. The study protocol was approved by the Bioethical Committee of the Silesian Medical Council (SIL/KB/662p/16).

### 2.2. Questionnaire

The questionnaire addressed socioeconomic data, an obstetric interview, preparation for delivery, self-reported pandemic anxiety questions, and participation in a birthing school. Validated tools were used to assess anxiety and depression, provision support, self-esteem, state and trait anxiety, delivery anxiety, life satisfaction, and marital communication before the delivery. In the postpartum part of the questionnaire, respondents were asked about labor, feelings and emotions during labor, and support from the partner during delivery. Additionally, those delivering during the COVID-19 pandemic were asked to answer queries concerning their subjective perception of COVID-19′s influence on pregnancy, delivery, and the relationship with the partner. 

#### 2.2.1. Brief Self-Rating Scale of Depression and Anxiety (BSRSDA) and Hospital Anxiety Depression Scale (HADS)

The Brief Self-Rating Scale of Depression was used to assess the level of anxiety and presence of depressive symptoms in peripartum. The scale is divided into two subscales consist of 10 questions, 5 of which ask about depression and 5 of which ask about anxiety. The intensity of essential symptoms was assessed via an eleven-point Likert scale. The symptoms include mood; sense of energy; intensity of interest; ability to experience pleasure; rate of thinking and acting; restlessness; mental tension; nervousness; anxiety; worry; feelings of physical tension; and desire to avoid situations. Scores between 0 and 4 are indicative of no symptoms, between 5 and 14 are interpreted as mild symptoms, whereas scores ≥15 are indicative of severe symptoms. The scale has good psychometric properties, with an α Cronbach score of −0.96 [17]. 

The HADS, containing 14 items, is commonly used in clinical practice to assess the intensity of negative emotions accompanying depression and anxiety. The result of the study is the sum of the points obtained, which was assessed separately for the anxiety scale and for the depression scale. A score equal to or greater than 11 points was taken as the cut-off point for anxiety and depressive disorders [18].

#### 2.2.2. Rosenberg Self-Esteem Scale (SES)

To evaluate global self-esteem, the Rosenberg Self-Esteem Scale (SES) in its Polish form was used [19]. The SES is a reliable research tool (the Cronbach’s alpha varies from 0.81 to 0.83) and consists of 10 statements assessed in a four-point scale [12]. Scores between 30 and 40 points indicate good self-esteem, between 26 and 29 points are considered average self-esteem, and 25 points or less is indicative of low self-esteem.

#### 2.2.3. The Satisfaction with Life Scale (SWLS)

The Satisfaction with Life Scale, designed by Diner et al., consists of 5 questions regarding the patient’s self-assessment of life, satisfaction with life, and achievement of life resolutions [20]. Each question was scored on a 7-point Likert scale, where 1 meant complete disagreement and 7 complete agreement with the statement. The result of the measurement is the overall level of satisfaction with life. The reliability of the Cronbach’s α scale = 0.82 [21]. The higher the score is, the greater the satisfaction with current life.

#### 2.2.4. The State-Trait Anxiety Inventory (STAI) 

Both the state and trait inventories consist of 20 statements, with a 4-point Likert scale from 1 (not at all) to 4 (very much so), and assesses general anxiety levels. Higher scores are indicative of higher levels of anxiety. The Cronbach’s α for state and trait anxiety are 0.93 and 0.94, respectively [22]. 

#### 2.2.5. Labor Anxiety Questionnaire (KLP II)

KLP II was used to assess the level of anxiety related to fear of labor. The scale consists of 9 items, which include attitudes toward labor and fear of labor. The answers are structured as statements to choose from: definitely yes, rather yes, rather no, definitely not. Individual statements are assigned numerical values from 0 to 3, according to the key. The statements 2, 3 and 5 hold values from 0 to 3, and the others (1, 4, 6, 7, 8, 9) hold values from 3 to 0. The scores vary from 0 to 27 points. The cut-off point for a low level of labor anxiety is 13 points. A score of 14–15 points indicates a slightly elevated level of labor anxiety, 16–17 means a high level of labor anxiety, while 18 points and above means very high labor anxiety. The scale is characterized by a satisfactory level of reliability (Cronbach’s alpha estimated at 0.69) [23].

#### 2.2.6. Berlin Social Support Scale (BSSS), and Provisions of Social Relation Scale (PSRS) from Partner

Emotional support was analyzed by the validated inventory of the Berlin Social Support Scale. The BSSS evaluates, in a multidimensional way, five aspects of social support (support available, the need for support, seeking support, received support and protection) using 17 items rated on a 4-point Likert scale. The higher the score is, the greater the level of social support. Satisfactory psychometric properties have been confirmed in previous Polish studies [24]. 

Support for a partner was assessed by the Provisions of Social Relation Scale. The questionnaire contained 8 questions regarding the level of support received from partners using a 4-point Likert scale, where 1 refers to “not true at all” and 5 “completely true”. Higher scores indicate a higher level of support. The scale shows good psychometric properties (Cronbach’s α, 0.91), and was previously used in a Polish population of women [24,25].

#### 2.2.7. Communication in Marriage Questionnaire (CMQ)

Communication processes in a marriage or relationship were assessed using the Communication in Marriage Questionnaire by M. Kazmierczak and M. Plopa. This research tool comprehensively measures the partners’ mutual communication behavior, including support, commitment, and appreciation. The questionnaire consists of two parts: each contains 30 statements regarding the self-assessment of communication and 30 statements regarding the assessment of the partner’s communication [26]. Higher scores indicate higher support, commitment, andappreciation. The scale shows good psychometric properties (Cronbach’s α, 0.70) and has been widely used in Poland [27]. 

Polish validated versions of all aforementioned scales were used.

### 2.3. Statistical Analysis

Statistical analysis was performed using SPPS 20.0 (IBM SPSS Statistics for Windows, Armonk, NY, USA: IBM Corp; 2012). As the analyzed variables were not normally distributed, and when comparing controls with cases, a non-parametric test was used (for qualitative variables—the Chi squared test, and for qualitative variables—the U-Mann–Whitney test). For evaluating the factors contributing to the level of delivery-related anxiety, a stepwise forward multiple regression model was used. In the first step, all variables were entered into the univariate model. For the final analysis, only those with statistical significance in the univariate analysis were included. The statistically significant *p* value was set at the level of <0.05.

## 3. Results

### 3.1. The General Characteristics of the Respondents

A total of 465 female participants completed the questionnaire in its entirety, including 190 before the pandemic and 275 during the COVID-19 pandemic. 

The mean age of respondents was 30.9 ± 4.9 years, and those in the COVID-19 group were older. Some differences between groups were noted. Women delivering during the COVID-19 pandemic had lower BMIs, and most of them resided in the city area. Primary and tertiary education levels were of statistical significance; the latter dominated in this group. Secondary education level did not differ between the groups. The vast majority of the women pregnant during COVID-19 pandemic were unemployed and, compared to the other group, fewer of them were black- and white-collar workers. In total, 98.9% of the women pregnant during the COVID-19 pandemic were in a relationship, which is statistically significant compared to the pre-pandemic group, and 100% of women pregnant during the pandemic had a history of previous deliveries, compared to 62.6% in another group. During the COVID-19 pandemic, fewer women (19.9%) had undergone an episiotomy, more women (37.8%) had an emergency C-section compared to the pre-pandemic period, and only 5.1% were administered an epidural in their previous delivery compared to pre-pandemic times. The number of previous pregnancies was higher (1.98 ± 1.14) in the COVID-19 pandemic group than in the other group (0.95 ± 1, *p* < 0.05). The characteristics of the respondents are presented in Table 1.

### 3.2. Brief Self-Rating Scale for Depression and Anxiety Scale (BSRSDA)

In the BSRSDA results, there were statistical differences in terms of both depression and anxiety, with higher scores (19.84 ± 13.23 for depression and 16.71 ± 12.53 for anxiety) in the group of women delivering during the COVID-19 pandemic. While the numbers of no (21.0%) and mild (56.2%) depression cases were higher before the pandemic, severe depression (66.7%) was much more prevalent during the COVID-19 pandemic. Mild anxiety was dominant in the first group (45.8%), whereas it was severe anxiety in the second group (49.6%).

### 3.3. Labor Anxiety Questionnaire (KLP II)

The KLP II showed no significant differences in the labor anxiety scores between the groups. However, taking the severity into consideration, moderate labor anxiety appeared more often (31.6%) before the pandemic, while severe levels were higher (20.6%) during the COVID-19 pandemic. There was also statistical significance in the no-anxiety group, showing higher scores (47.1%) during the pandemic.

### 3.4. Hospital Anxiety and Depression Scale—HADS

The HADS anxiety scores were lower (7.32 ± 4.53) in the group of pregnant women during the COVID-19 pandemic compared to the group of women who were pregnant before the pandemic (7.75 ± 3.28). The HADS depression scores showed no statistical significance, whereas the HADS depressive symptom scores (11.6%) and anxiety symptom scores (21.0%) were higher in women delivering during the pandemic. 

### 3.5. Communication in Marriage Questionnaire (CMQ)

In the group of women pregnant during the COVID-19 pandemic, support for the partner and commitment to the partner were statistically significantly higher compared to the women delivering before the COVID-19 pandemic. The rest of the CMQ components (appreciation of perceived support from partner, perceived commitment with partner and perceived partner’s appreciation) were not statistically significant. 

### 3.6. Factors Influencing Delivery-Related Anxiety

To verify which factors may influence the level of anxiety related to fear of delivery, a regression model was used with the Brief Self-Rating Scale of Depression and Anxiety and the Self-Report Labor Anxiety Questionnaire as dependent variables. The analysis reveals that anxiety measured by BSRSDA was higher in those with depressive symptoms, according to HADS-D (t = 6.2, β = 0.28, *p* = 0.0001), with a higher level of trait anxiety according to STAI-2 (t = 9.6, β = 0.44, *p* = 0.0001) that was lower in those delivering before the COVID-19 pandemic (t = −4.0, β = −0.17, *p* = 0.000). This model presents satisfactory parameters—F(338;3) = 66.7, *p* = 0.001, R2 = 0.37. However, when using the Self-Report Labor Anxiety Questionnaire as the dependent variable, women with lower self-esteem according to SES (t = −3.65, β = −0.17, *p* = 0.000), a higher level of trait anxiety according to STAI-2 (t = 7.3, β = 0.33, *p* = 0.0001), and who are white-collar workers (t = 2.59, β = 0.16, *p* = 0.0001) had a higher level of labor-related anxiety—F(407;4) = 20.2, *p* = 0.001, R2 = 0.21. Similarly, when the HADS anxiety scale was used, a higher level of anxiety was related to the presence of depressive symptoms (t = 4.43, β = 0.27, *p* = 0.0001), the level of trait anxiety according to STAI-2 (t = 7.2, β = 0.43, *p* = 0.0001), and not having had an epidural in the previous pregnancy (t = −2.7, β = −0.53, *p* = 0.0001)—F(203;3) = 35.4, *p* = 0.001, R2 = 0.34. Finally, when STAI-1 was used as the indicator of anxiety, the presence of depressive symptoms (t = 2.93, β = 0.18, *p* = 0.001), level of trait anxiety according to STAI-2 (t = 6.52, β = 0.28, *p* = 0.001), level of perceived support from partner according to CMQ (t = −3.1, β = −0.19, *p* = 0.001), and not having had an epidural in the previous pregnancy (t = −2.37, β = 0.14, *p* = 0.01) were risk factors for higher anxiety—F(193;4) = 27.2, *p* = 0.01, R2 = 0.36.

## 4. Discussion

### 4.1. General Remarks—COVID-19 Pandemics, Depression and Trait Anxiety as a Risk Factor for Anxiety Related to Delivery

Pregnancy is a unique period of life, in which biological, physiological, and psychological changes occur [28]. Mental ambivalence, mood changes, and the occurrence of emotional or anxiety–depressive disorders are observed [1,2,3,29]. The mother’s fears about the course of pregnancy and childbirth may cause stress that affects the psychological and emotional quality of life of women and their partners [28,29,30,31]. As psychiatric disorders, including depression and anxiety, have been common among pregnant women during the COVID-19 pandemic [17], the objective of our study was to determine the impact of the COVID-19 pandemic on anxiety related to labor in pregnant and postpartum women. In addition, factors associated with an increased risk of depressive symptoms, anxieties, and perinatal anxiety were analyzed. The analysis of the results indicates that the COVID-19 pandemic has influenced the perception of fear of labor and depression/anxiety related to labor. However, other factors such as partner’s support or history of epidural during previous delivery also have an impact on state and trait anxiety levels during pregnancy. 

The results of our study have shown, as hypothasized, that severe anxiety, as assessed by the Brief Self-Rating Scale of Depression and Anxiety, was observed in 49.6% of pregnant women during the COVID-19 pandemic, compared to 32% in pregnancy before the COVID-19 pandemic. A similar result was reported in the resent paper by Nowacka et al. where 48% of women with COVID-19 infection had a clinically significant level of anxiety [31]. This higher level might be explained by concern about the health of the fetus, the possibility of vertical infection during delivery or infection of the infant [32], fear of possible developmental anomalies in the fetus resulting from SARS-CoV-2 infection, fetal growth restriction [33], concerns about inadequate care during pregnancy, or concerns due to social restriction and social avoidance due to infection risk [31,34]. Similarly, when the Hospital Anxiety and Depression Scale was used, the prevalence of depressive symptoms was significantly higher during the COVID-19 pandemic (11.6% vs. 6.3%). This is in line with the results of the study conducted by King et al., which revealed that pregnant women during the pandemic were twice as likely to suffer from depression [35]. Similarly to anxiety, some factors might increase the risk of depression in pregnancy; these are believed to be the physical and mental condition of the mother, earlier maternal experiences, relationship and epidemiological circumstances [36], lower education level, unemployment, reduction in perception of general support, and financial problems [14]. Interestingly, recent studies, including our own, have shown that the COVID-19 pandemic may increase the prevalence of perinatal and postpartum depression [37]. Additionally, as noted by Grumi et al., emotional stress and partial social support during the COVID-19 pandemic can contribute to developing mental issues, such as depression and anxiety [38]. The level of depression might also fluctuate during different trimesters of pregnancy, with the highest levels in the third [39] or the first [40]. However, we did not assess these levels according to pregnancy trimesters. Based on the results of Kahyaoglu’s study, the severity of perinatal anxiety is not correlated with the trimester of pregnancy [41].

### 4.2. Factors Affecting Perinatal Anxiety

Some important factors that impact perinatal anxiety levels have been observed in our study. As expected, regardless of the scale used to assess the anxiety level, being pregnant during COVID-19, the presence of depressive symptoms, trait anxiety level (discussed in the previous paragraph), self-esteem, and not receiving an epidural in the previous pregnancy increased the anxiety level related to delivery. Surprisingly, higher education level was also a risk factor. The hypothesis on the correlation between perceived support from the partner was only partially confirmed. 

#### 4.2.1. Partners Support

Although the support received from the partner, objectively measured via the BSSS and PSRS, did not differ between groups, the lack of that support was a risk factor for higher anxiety, as assessed by STAI-1. Our results are confirmed by previous observations that social support may be one of the main factors affecting both mental and physical health [42,43]. As studies have shown, there is a significant correlation between the amount of perceived support from the partner and depression or anxiety level, especially among pregnant women. According to the research by Iwanowicz-Palus et al., support from loved ones was significant in achieving a good psychological condition and decreasing anxiety during childbirth in the COVID-19 epidemiological situation [36]. Furthermore, in their study, Zhang et al. emphasized the link between stress derived from lifestyle restrictions or insufficient emotional and instrumental support during pregnancy, and the hypersecretion of cortisol, which is considered to be a biologic risk factor for depression [14].

#### 4.2.2. Education Level

The results of our study show that white collar workers had higher levels of labor anxiety compared to other women. This is in contrast with the results of the study by Çolak et al., wherein a low educational level and a low income were listed as factors increasing the anxiety levels in pregnant women [39]. Similar conclusions were drawn in a study by Luo et al., wherein unemployment, a lower education and financial issues were risk factors for anxiety [14]. In the paper by Yadav et al., financial problems due to a lack of education were related to anxiety in pregnant women during the COVID-19 pandemic [44]. Similarly, lower education was correlated with anxiety due to cesarean delivery in the recently published study by Ferede et al. [45]. It might be speculated that higher education is related to greater knowledge of COVID-19′s consequences for the fetus, which might increase the fear of labor [17]. However, this has yet to be confirmed in other studies evaluating that knowledge. 

#### 4.2.3. Previous Pregnancies and Epidural

The results of this study show that women during the pandemic presented a higher number of pregnancies and deliveries compared to pre-pandemic women. In addition, the women giving birth during the COVID 19 pandemic were characterized by higher levels of perinatal anxiety. This is in contrast with the findings from research conducted by Farewell et al. These authors have described higher levels of perinatal anxiety among women who were pregnant for the first time. They explained their results via the tendency to fear unknown events, or beliefs about the need for emergency medical intervention during labor [36]. On the other hand, a higher level of anxiety in multiparous women might be related to being pregnant during the COVID-19 pandemic, when access to medical services was limited and the fear of the perinatal transition of COVID-19 was high [17]. These factors might augment the anxiety level. However, multiparty was not a risk factor for higher perinatal anxiety. Additionally, our study has confirmed that effective pain management (epidural) during the previous pregnancy is a protective factor for anxiety during the next pregnancy. This is in line with results of other studies, showing that the possibility of pain reduction during labor reduces the stress related to delivery [46,47]. 

#### 4.2.4. Self-Esteem

Although in our study, both groups had low self-esteem during pregnancy, slightly higher scores were noted during the COVID-19 pandemic. According to the study by Van Scheppingen et al., the self-esteem of future mothers fluctuates with trimesters and postpartum, with reductions in the second, followed by an increase around childbirth and a gradual reduction in the following years [15]. This may vary somewhat due to the initial level of self-esteem in mothers, the satisfaction with the romantic relationship, the partner’s attitude towards gaining weight, and education level and paid employment [15,48].

The higher level of self-esteem in those delivering during the COVID-19 pandemic, as seen in our study, might be at least partially explained by the lower BMI level in this group. Additionally, lower self-esteem was correlated with higher anxiety. This is in line with the results of a recent study by Han et al., showing that depression, anxiety and self-esteem in postnatal women were affected by weight gain during pregnancy, as well as the concerns about not being able to return to pre-pregnancy weights [49]. Similarly, in the study by Cevik and Yanikkerem, positive body image correlates with the prevalence of anxiety symptoms and depressive symptoms in pregnancy [16].

### 4.3. Strengths and Limitations

To our knowledge, this is one of few studies directly comparing peripartum anxiety before COVID-19 and during the pandemic, showing that COVID-19 has a direct effect on anxiety levels. We managed to identify some modifiable factors that can influence the peripartum anxiety level. From the clinical point of view, as other COVID-19 waves are expected, actions to reduce this anxiety should be taken in order to improve the mental health of pregnant women. Psychological support using telemedicine might be helpful when consulting with patients with anxiety and depressive symptoms during COVID-19, if face-to-face intervention is impossible [50]. It might also be suggested that clinicians screen for trait anxiety, lack of family/partner support, depressive symptoms, and low self-esteem. The use of appropriate scale (STAI, Rosenberg Self-Esteem Scale, Communication in Marriage Questionnaire) and multidisciplinary consultations as a triage—with the aim of identifying the susceptible population—could contribute to a decrease in peripartum anxiety levels, especially in those who did not have an epidural in previous pregnancies and with higher education level.

This study also has some limitations. It was designed primarily to assess the patient’s current mental state through a questionnaire. We only have data from wave three of the pandemic on the severity of the above symptoms, and these symptoms may have varied, depending on the duration of the pandemic, among pregnant women. However, as the number of COVID-19 cases in the general population and in pregnant women did not differ significantly between the third and second waves, this will not have biased the results. In addition, only women in the third trimester of pregnancy completed the questionnaire, though the course and severity of the psychopathological symptoms may have fluctuated during the whole pregnancy period. It is also worth noting that the presence of various restrictions, which changed rapidly, could have affected the mental health of pregnant women in Poland. Finally, in the COVID-19 group, all women were multiparous, so the results cannot be generalized to the whole population of pregnant women in Poland. However, we consider that none of these factors have significantly influenced the results of the study. 

## 5. Conclusions

The COVID-19 pandemic increases the prevalence of peripartum anxiety related to pregnancy/labor, depressive symptoms in pregnancy and the support given to the partner and commitment to the partner. Being pregnant and delivering during the COVID-19 pandemic is a risk factor for peripartum anxiety. However, lower self-esteem, high level of trait anxiety, presence of depressive symptoms, poor pain management in previous deliveries, and low support received from a partner are independent factors exaggerating anxiety levels during pregnancy and labor. All these factors should be considered when counseling couples before the delivery to decrease the peripartum anxiety.

## Figures and Tables

**Table 1 jcm-11-06522-t001:** Socio-demographic and psychological characteristics of study participants.

Factor		Pregnancy before COVID-19 Pandemic Mean %	Pregnancy during COVID-19 Pandemic Mean %	Total	*p*
Age		29.71 ± 4.52 (19–44)	31.32 ± 4.67 (18–45)	30.87 ± 4.94 (18–45)	0.001
BMI		27.64 ± 5.1 (17–64.3)	26.9 ± 5.1 (17–42)	27.2 ± 5.12 (17–64)	0.03
Residency	Rural	61.9% (60)	38.1% (37)	20.8% (97)	0.001
	City	35.2% (130)	64.8% (239)	79.2% (369)	
Education level	Primary	56.8% (21)	43.2% (16)	7.9% (37)	0.02
	Secondary	50.0% (69)	50.0% (69)	29.6% (138)	0.98
	Tertiary	34.4% (100)	65.6% (191)	62.4% (291)	0.001
Employment	Black collar	28.9% (55)	5.8% (16)	15.2% (71)	0.001
	White collar	49.5% (94)	25.0% (69)	35.0% (163)	0.001
	No	21.6% (41)	69.2% (191)	49.8% (232)	0.001
Relationship (Yes)		92.6% (176)	98.9% (273)	96.3% (449)	0.001
Duration of RS (years)		7.98 ± 5.1 (0–25)	8.82 ± 4.45 (1–24)	8.59 ± 4.55 (0–25)	0.09
RS quality #		4.6 ± 0.77 (1–6)	4.59 ± 0.69 (2–5)	4.59 ± 0.72 (1–6)	0.85
Planned pregnancy (Yes)		80.5% (153)	84.8% (234)	83.0% (387)	0.22
No. of previous pregnancies		0.95 ± 1 (0–5)	1.98 ± 1.14 (0–7)	1.56 ± 1.2 (0–7)	0.0001
No. of previous deliveries		0.74 ± 0.79 (0–3)	1.63 ± 0.81 (0–6)	1.26 ± 0.91 (0–6)	0.0001
Previous deliveries	Without episiotomy	10.5% (20)	15.9% (44)	13.7% (64)	0.09
	With episiotomy	32.6% (62)	19.9% (55)	25.1% (117)	0.03
	Instrumental delivery	3.2% (6)	2.5% (7)	2.8% (13)	0.91
	Emergency C-section	6.8% (13)	37.8% (104)	25.1% (117)	0.001
	With epidural	48.9% (46)	5.1% (7)	22.8% (53)	0.001
	Family delivery	66.4% (75)	65.6% (181)	65.8% (256)	0.88
	Complications in previous delivery	15.7% (17)	23.9% (66)	21.6% (83)	0.08
	Total	62.6% (119)	100% (275)	84.8% (395)	0.001
No. of previous miscarriages		0.25 ± 0.51 (0–3)	0.37 ± 0.74 (0–7)	0.32 ± 0.66 (0–7)	0.28
Socioeconomic status #		3.68± 0.74 (1–5)	3.92 ± 0.69 (2–5)	3.82 ± 0.72 (1–5)	0.001
BSRSDA—Depression		8.21 ± 7.38 (0–36)	19.84 ± 13.23 (0–50)	15.1 ± 12.58 (0–50)	0.0001
BSRSDA—Anxiety		11.67 ± 9.23 (0–42)	16.71 ± 12.53 (0–50)	14.66 ± 11.56 (0–50)	0.0001
BSRSDA—Depression severity	None	21.0% (40)	11.2% (31)	15.2% (71)	0.001
	Mild	56.2% (107)	22.1% (61)	36.0% (168)	0.001
	Severe	22.6% (43)	66.7% (184)	48.8% (227)	0.001
BSRSDA L—Anxiety severity	None	22.1% (42)	21.0% (58)	21.5% (100)	0.21
	Mild	45.8% (87)	29.3% (81)	36.0% (168)	0.001
	Severe	32.1% (61)	49.6% (137)	42.5% (198)	0.001
KLP II—peripartum anxiety score		14.53 ± 3.2 (6–25)	13.98 ± 4.04 (4–24)	14.16 ± 3.73 (4025)	0.09
KLP II—severity	None	34.2% (65)	47.1% (130)	41.8% (195)	0.01
	Mild	22.1% (42)	17.4% (48)	19.3% (90)	0.24
	Moderate	31.6% (60)	14.9% (41)	21.7% (101)	0.001
	Severe	12.1% (23)	20.6% (57)	17.2% (80)	0.02
STAI-I		37.24 ± 8.6 (22–68)	38.8 ± 10.91 (20–74)	38.17 ± 10.06 (20–74)	0.16
STAI-II		39.15 ± 7.89 (24–58)	39.05 ± 8.67 (21–68)	39.1 ± 8.36 (21–68)	0.96
HADS depression		4.74 ± 3.35 (1–14)	5.14 ± 3.85 (0–19)	4.98 ± 3.66 (0–19)	0.45
HADS anxiety		7.75 ± 3.28 (1–19)	7.32 ± 4.53 (0–20)	7.5 ± 4.07 (0–20)	0.03
HADS—depressive symptoms		6.3% (12)	11.6% (32)	9.4% (44)	0.03
HADS—anxiety symptoms		20.5% (39)	21.0% (58)	20.8% (97)	0.89
SWLS		24.1 ± 4.67 (9–34)	24.6 ± 5.7 (5–35)	24.4 ± 5.31 (5–35)	0.87
BSSS		29.74 ± 2.94 (17–32)	29.06 ± 4.24 (8–32)	29.33 ± 3.77 (8–32)	0.79
PSRS		33.84 ± 6.61 (5–40)	34.36 ± 6.67 (15–40)	34.15 ± 6.67 (5–40)	0.07
SES		19.21 ± 6.02 (10–40)	22.96 ± 2.2 (17–30)	21.73 ± 4.45 (10–40)	0.00001
CMQ support to partner		44.21 ± 4.55 (28–50)	44.66 ± 5.73 (10–50)	44.48 ± 5.28 (10–50)	0.03
CMQ commitment to partner		34.66 ± 4.67 (18–44)	36.66 ± 5.52 (9–45)	35.83 ± 5.27 (9–45)	0.0001
CMQ appreciation of partner		19.79 ± 5.77 (14–50)	20.91 ± 6.92 (11–55)	20.45 ± 6.49 (11–55)	0.13
CMQ perceived support from partner		42.9 ± 5.9 (14–50)	42.84 ± 7.68 (10–50)	42.86 ± 6.98 (10–50)	0.15
CMQ perceived commitment to partner		34.43 ± 5.75 (16–45)	34.6 ± 6.96 (9–45)	34.53 ± 6.48 (9–45)	0.38
CMQ perceived partner’s appreciation		17 ± 5.46 (11–44)	18.1 ± 6.97 (11–55)	17.65 ± 6.41 (11–55)	0.25

BMI—Body Mass Index; KLP II—Labor Anxiety Questionnaire; BSRSDA—Brief Self-Rating Scale of Depression and Anxiety; HADS—Anxiety and Hospital Anxiety Depression Scale; SES—Rosenberg Self-Esteem Scale; SWLS—Satisfaction with Life Scale; STAI—State–Trait Anxiety Inventory; BSSS—Berlin Social Support Scale; PSRS—Provisions of Social Relation Scale (from partner); CMQ—Communication in Marriage Questionnaire; RS—relationship; #—5-point Likert scale.

## Data Availability

The quantitative datasets supporting the conclusions of this article are included in the article. More detailed datasets are available from the corresponding author on request.

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
