# Peer review of "Does the COVID-19 Pandemic Affect Labor-Related Anxiety and Prevalence of Depressive Symptoms in Pregnant Women?"

_jcm, 2022, doi:10.3390/jcm11216522_

Round 1
Reviewer 1 Report (New Reviewer)
Thank you for the opportunity to review this paper. The topic of the research and data obtained are undoubtly valuable.
I suggest to clarify the aims of the study and to bring them into line with the title of the paper. As far as I understand, this work is dedicated to the labor related anxity and depression. The fear of childbirth is the separate phenomenon, which was not investigated in the study, but is indicated in the title.
In the Results, Discussion and Conclusion section I would suggest to concentraite on the aims of the study, specificly on the labor related anxiety and depression and their risk factors. The describton of the vast amount of variables semms to overload the paper. Maybe tthe data obtained could be devided into two separate papers? Some of the variables could be analysed as the covariates. Or there might be a more claer and structured model, describing the results of the study.
Author Response
Reviewer 1.
Dear Reviewer,
Thank you very much for your valuable remarks. The answers are listed below.
- I suggest to clarify the aims of the study and to bring them into line with the title of the paper. As far as I understand, this work is dedicated to the labor related anxity and depression. The fear of childbirth is the separate phenomenon, which was not investigated in the study, but is indicated in the title.
Ad 1. As suggested, the title of the paper was changed to “Does the COVID-19 pandemic affects labor-related anxiety and prevalence of depressive symptoms in pregnant women?”
- In the Results, Discussion and Conclusion section I would suggest to concentraite on the aims of the study, specificly on the labor related anxiety and depression and their risk factors. The describton of the vast amount of variables semms to overload the paper. Maybe tthe data obtained could be devided into two separate papers? Some of the variables could be analysed as the covariates. Or there might be a more claer and structured model, describing the results of the study.
Ad 2. As suggested, those sections were changed to be concentrated on depression and anxiety. To be more specific, the section happiness to the pregnancy was deleted. However, other sections seems to be important especially that they are correlated with factors affecting anxiety and depression related to labor.
Reviewer 2 Report (New Reviewer)
The authors conducted a study to explore the impact of COVID-19 on Poland's pregnant women's mental health and associated moderating factors in this relationship. It would be great to see the Poland evidence published and I am interested in this article. However, I think this article needs to be greatly improved in order to bring more value after publication. In other words, it is not appropriate to be published in its current form. Below I have outlined my major comments about the manuscript, which I hope will help improve it further.
In the "Abstract" part, you need to indicate the meaning of the data presented. For example, what is the meaning of "1.30 vs 1.44"? Is it mean value, median value, adjusted OR, or RR? Although after looking through your main text, the reader could find the meaning of these figures, you still need to clearly present it in the "Abstract".
It is recommended to proofread and revise your whole paper before submitting it to the journal. There are some sentences in the text that are not very clearly stated and may confuse the reader. For example, in the first sentence of "Introduction", although the data is from WHO, what is the scope of the data, and is this global prevalence data? You know, WHO also has data for a country or a region, rather than referring to global data when WHO is mentioned. In the whole text, there are other unclear sentences.
It is suggested to add more references in the section 2.1 "Study design and participants". For example, why do you choose the inclusion and exclusion criteria that you've shown here? If you could show some evidence from existing literatures, it would be better. Moreover, it is very good to conduct sample size calculations, but why is the "difference in the prevalence of severe anxiety based on the Brief Self-Rating Scale of Depression and Anxiety between cases and controls" 10% and the "difference in the prevalence of severe peripartum anxiety based on Labor Anxiety Questionnaire scores" 8%? Did you reference other studies or just based on your experiences?
If it is a "cohort study", as you mentioned in the "Abstract", how did you collect data of the control group? Because you mentioned the control group is those who gave birth before the COVID-19 pandemic, but you also mentioned the questionnaire "was given to participants twice: up to 7 days before the delivery and one day postpartum". Did it mean that the data for the exposure group and control group were collected at different times? If so, you should present it.
I guess a table may be forgotten to be presented in your manuscript? You show some figures in section 3.6, "Factors influencing delivery-related anxiety", but it would be better if you could also present your result in a table.
Maybe you finished your manuscript early, but now it is not the "one of the first studies comparing peripartum anxiety before COVID-19 and during the pandemic". I recommended compare your study with others conducted in Europe or in Poland, if you still want to presnet sentence like it.
It is suggested to show the representativeness of your sample for women in Poland. Based on the characteristics of your sample, such as "100% of women pregnant during the pandemic had a history of previous deliveries", only collected from "at the Department of Obstetrics and Gynecology of the University Clinical Center of the Medical University of Silesia in Katowice", etc. You may need to justify the generality of your sample to the whole population, since one of your aims is to "determine the prevalence of symptoms of anxiety and depression during pregnancy in a population of Polish women", otherwise it would be better if you could add some sentence about the representativeness problem of your study.
Author Response
Reviewer 2
The authors conducted a study to explore the impact of COVID-19 on Poland's pregnant women's mental health and associated moderating factors in this relationship. It would be great to see the Poland evidence published and I am interested in this article. However, I think this article needs to be greatly improved in order to bring more value after publication. In other words, it is not appropriate to be published in its current form. Below I have outlined my major comments about the manuscript, which I hope will help improve it further.
Dear Reviewer,
Thank you very much for your valuable remarks. The answers are listed below.
- In the "Abstract" part, you need to indicate the meaning of the data presented. For example, what is the meaning of "1.30 vs 1.44"? Is it mean value, median value, adjusted OR, or RR? Although after looking through your main text, the reader could find the meaning of these figures, you still need to clearly present it in the "Abstract".
Ad 1. That data was changed to meet the reviewer exceptions.
- It is recommended to proofread and revise your whole paper before submitting it to the journal. There are some sentences in the text that are not very clearly stated and may confuse the reader. For example, in the first sentence of "Introduction", although the data is from WHO, what is the scope of the data, and is this global prevalence data? You know, WHO also has data for a country or a region, rather than referring to global data when WHO is mentioned. In the whole text, there are other unclear sentences.
Ad 2. As suggested the text was revised to correct the confusing fragments
- It is suggested to add more references in the section 2.1 "Study design and participants". For example, why do you choose the inclusion and exclusion criteria that you've shown here? If you could show some evidence from existing literatures, it would be better. Moreover, it is very good to conduct sample size calculations, but why is the "difference in the prevalence of severe anxiety based on the Brief Self-Rating Scale of Depression and Anxiety between cases and controls" 10% and the "difference in the prevalence of severe peripartum anxiety based on Labor Anxiety Questionnaire scores" 8%? Did you reference other studies or just based on your experiences?
Ad 3. All exclusion and inclusion criteria were bested on literature search. As high-risk pregnancy is a risk factor for depression and anxiety, only women with physiological uncomplicated pregnancies were included (that information in that was added).
As for sample size calculation we based our calculation on the data form previous studies – the pre-covid level of depression and anxiety was 8-12% and 13-22% respectively [Mazurkiewicz, D.W.; Strzelecka, J.; Piechocka, D.I. Adverse Mental Health Sequelae of COVID-19 Pandemic in the Pregnant Population and Useful Implications for Clinical Practice. J Clin Med 2022, 11, doi:10.3390/jcm11082072. Puertas-Gonzalez, J.A.; Mariño-Narvaez, C.; Peralta-Ramirez, M.I.; Romero-Gonzalez, B. The psychological impact of the COVID-19 pandemic on pregnant women. Psychiatry Res 2021, 301, 113978, doi:10.1016/j.psychres.2021.113978.]. The prevalence of during Covid was between 25 and 37% for depression and 43-71% for anxiety [Mazurkiewicz, D.W.; Strzelecka, J.; Piechocka, D.I. Adverse Mental Health Sequelae of COVID-19 Pandemic in the Pregnant Population and Useful Implications for Clinical Practice. J Clin Med 2022, 11, doi:10.3390/jcm11082072.]. As the frequency of severe symptoms in lower, we decided to set the difference to be at least 10 and 8%, respectively. That information was added to the methods section.
- If it is a "cohort study", as you mentioned in the "Abstract", how did you collect data of the control group? Because you mentioned the control group is those who gave birth before the COVID-19 pandemic, but you also mentioned the questionnaire "was given to participants twice: up to 7 days before the delivery and one day postpartum". Did it mean that the data for the exposure group and control group were collected at different times? If so, you should present it.
Ad 4. That is current. The data for the control group was collected earlier. That was clearly stated in the Methods section (“In total, 1664 women delivering at the Department of Obstetrics and Gynecology of the University Clinical Center of the Medical University of Silesia in Katowice before the COVID-19 pandemic, and 1748 delivering during the pandemic were eligible for this questionnaire-based cohort study”). As it is impossible to present all the information in the abstract, we decided not to add any new data to that part of MS
- I guess a table may be forgotten to be presented in your manuscript? You show some figures in section 3.6, "Factors influencing delivery-related anxiety", but it would be better if you could also present your result in a table.
Ad 5. Thank you for that suggestion. However, that data, in our opinion, would be illegible if was presented in the table.
- Maybe you finished your manuscript early, but now it is not the "one of the first studies comparing peripartum anxiety before COVID-19 and during the pandemic". I recommended compare your study with others conducted in Europe or in Poland, if you still want to presnet sentence like it.
Ad 6. Thank you for that suggestion. The sentence was changed to “one of few studies comparing directly”
- It is suggested to show the representativeness of your sample for women in Poland. Based on the characteristics of your sample, such as "100% of women pregnant during the pandemic had a history of previous deliveries", only collected from "at the Department of Obstetrics and Gynecology of the University Clinical Center of the Medical University of Silesia in Katowice", etc. You may need to justify the generality of your sample to the whole population, since one of your aims is to "determine the prevalence of symptoms of anxiety and depression during pregnancy in a population of Polish women", otherwise it would be better if you could add some sentence about the representativeness problem of your study.
Ad 7. Thank you for that suggestion. As indeed the sample in not representative for whole population in Poland, we changed the aims and add the information that the results cannot be generalized to all pregnant women in Poland.
Reviewer 3 Report (New Reviewer)
Manuscript jcm-1948971, titled “Does the COVID-19 pandemic affect pregnant women's fear of labor?” was submitted to Journal of Clinical Medicine for possible publication. It appears that this may have been a revision of a prior draft. This paper is an empirical examination of the differences in anxiety and depression as well as fear of labor in women who were pregnant prior to the COVID pandemic and those pregnant during the COVID pandemic. The authors further looked at potential sociodemographic risk factors for the anxiety and depression symptoms in this population. Results indicated that women pregnant during COVID had greater levels of anxiety, depression, and fear of labor, and lower happiness about pregnancy compared to those pregnant before COVID. Several significant risk factors were identified.
Overall, this manuscript has the potential to provide an important contribution to the literature, as the effects of the pandemic on mental health during pregnancy is an important topic. Strengths of the manuscript include use of multiple validated measures and relatively large sample sizes. The statistics generally seem appropriate. At a broad level, I think the authors could bolster the support for the novelty of their study as well as enhance the discussion by providing a more in-depth discussion of the applications of their results. I list below some specific suggestions for the authors for future revision.
Introduction:
-The authors noted that pre-COVID, the rate of anxiety during pregnancy was 22%. They then mentioned that it was 23% during COVID. Are those numbers correct? It seemed that the focus of that paragraph was on the increased risk for mental illness during COVID, but it’s unclear if a 1% increase is clinically meaningful. It would be helpful if the authors could clarify this.
-It would be useful if the authors could be more specific when describing the aims of the study. For instance, they could list the variables they chose to measure (especially the risk factors) and why. Did the authors have any specific hypotheses? If so, those should be listed too.
Method:
-The authors stated that the Brief Self-Rating Scale of Depression refers to fear of delivery but they also say it measures anxiety and depression. Can the authors clarify if the questions are asking about anxiety and depression in relation to “fear of delivery” specifically?
-It would be useful to state the number of items on the depression and anxiety measures
Results:
-The table is a very useful visual for presenting the results
-It seems that for the regression analyses involving the psychosocial predictor variables, the analysis was run on the entire sample, is that correct? Have the authors considered running the regression analyses separately by sample to see if the predictors differ in their two subgroups? I wonder if some differences could have been masked by combining the sample. If combining the sample is the better option, perhaps the authors can give a rationale for why they did it this way.
Discussion:
-The Discussion was very brief. This part could be strengtheed with more of the authors’ ideas about the applicability of their results. They mentioned the need to put in preventive measures to decrease risk for anxiety and depression in this population during the pandemic. Since they measured many variables that were found to be significant predictors of anxiety and depression, I wonder if they could perhaps provide some thoughts on how providers could leverage knowledge about these other variables to identify the women most at risk for symptoms, and work to prevent/decrease these symptoms in pregnant women.
-It would also be useful to state clearly what novel knowledge is gained in this study.
Minor:
- line 113: there is an extra “Anxiety” in the sentence
-line 207: “compared” is misspelled
Author Response
Reviewer 3
Manuscript jcm-1948971, titled “Does the COVID-19 pandemic affect pregnant women's fear of labor?” was submitted to Journal of Clinical Medicine for possible publication. It appears that this may have been a revision of a prior draft. This paper is an empirical examination of the differences in anxiety and depression as well as fear of labor in women who were pregnant prior to the COVID pandemic and those pregnant during the COVID pandemic. The authors further looked at potential sociodemographic risk factors for the anxiety and depression symptoms in this population. Results indicated that women pregnant during COVID had greater levels of anxiety, depression, and fear of labor, and lower happiness about pregnancy compared to those pregnant before COVID. Several significant risk factors were identified.
Overall, this manuscript has the potential to provide an important contribution to the literature, as the effects of the pandemic on mental health during pregnancy is an important topic. Strengths of the manuscript include use of multiple validated measures and relatively large sample sizes. The statistics generally seem appropriate. At a broad level, I think the authors could bolster the support for the novelty of their study as well as enhance the discussion by providing a more in-depth discussion of the applications of their results. I list below some specific suggestions for the authors for future revision
Dear reviewer,
Thank you very much for your valuable remarks. The answers are listed below.
- Introduction:
The authors noted that pre-COVID, the rate of anxiety during pregnancy was 22%. They then mentioned that it was 23% during COVID. Are those numbers correct? It seemed that the focus of that paragraph was on the increased risk for mental illness during COVID, but it’s unclear if a 1% increase is clinically meaningful. It would be helpful if the authors could clarify this.
Ad 1. We checked the data – the prevalence of anxiety during COVID-19 pandemic varies between 43 and 71%. That information was corrected
- It would be useful if the authors could be more specific when describing the aims of the study. For instance, they could list the variables they chose to measure (especially the risk factors) and why. Did the authors have any specific hypotheses? If so, those should be listed too.
Ad 2. We based our assumptions on the results of previous studies. We added some hypothesis to aims of the study.
- Method:
The authors stated that the Brief Self-Rating Scale of Depression refers to fear of delivery but they also say it measures anxiety and depression. Can the authors clarify if the questions are asking about anxiety and depression in relation to “fear of delivery” specifically?
Ad 3. Thank you for that suggestion. We change the sentence to “The Brief Self-Rating Scale of Depression was used to assess anxiety level and presence of depressive symptoms in peripartum” to correspond with what the scale was used for.
- it would be useful to state the number of items on the depression and anxiety measures
Ad 5. Added as requested.
- Results:
The table is a very useful visual for presenting the results
Ad 6. Thank you for that remark.
- It seems that for the regression analyses involving the psychosocial predictor variables, the analysis was run on the entire sample, is that correct? Have the authors considered running the regression analyses separately by sample to see if the predictors differ in their two subgroups? I wonder if some differences could have been masked by combining the sample. If combining the sample is the better option, perhaps the authors can give a rationale for why they did it this way.
Ad 7. Indeed, we perform the analysis on the entire sample to verify in being pregnant during COVID-19 pandemic is a risk factor for labor-related anxiety. As adding additional analysis could be confusing and as the aim of the study was to check if COVID-19 pandemic itself is related to increase anxiety, we decided not to add another analysis. However, as the reviewer suggested, that could be an interesting subject for another paper.
- Discussion:
The Discussion was very brief. This part could be strengtheed with more of the authors’ ideas about the applicability of their results. They mentioned the need to put in preventive measures to decrease risk for anxiety and depression in this population during the pandemic. Since they measured many variables that were found to be significant predictors of anxiety and depression, I wonder if they could perhaps provide some thoughts on how providers could leverage knowledge about these other variables to identify the women most at risk for symptoms, and work to prevent/decrease these symptoms in pregnant women.
Ad 8. As suggested we added some ides in the Strengths and limitations Section of the manuscript: “It might also be suggested for clinician to screen for trait anxiety, lack of family/partner support, depressive symptoms, and low self-esteem. Using appropriate scale (STAI, Rosenberg Self-Esteem Scale, Communication in Marriage Questionnaire) and multidisciplinary consultations as a triage aimed to identify susceptible population could contribute to decrease in peripartum anxiety level, especially in those who did not have epidural in previous pregnancies and with higher education level”.
- It would also be useful to state clearly what novel knowledge is gained in this study.
Ad. 9 We added the sentence “We managed to identify some modifiable factors that can influence the peripartum anxiety level”
10. Minor:
- line 113: there is an extra “Anxiety” in the sentence
-line 207: “compared” is misspelled
Ad 10. Corrected
Round 2
Reviewer 2 Report (New Reviewer)
No further comments.
This manuscript is a resubmission of an earlier submission. The following is a list of the peer review reports and author responses from that submission.
Round 1
Reviewer 1 Report
Regarding the structure and accuracy of the phrases, unfortunatelly the manuscript lacks in well structured information, and the phrases are not so well designed. Being an, observational retrospective questionary-based analysis, the study followed a group of 465 women, divided in 2 groups, but this may not be sufficient for such a study, as the 2 groups were analysed only in the period of the third wave of COVID-19. The aim of the analysis was to assess the impact of the COVID-19 Pandemic on adult pregnant women and also the post-partum women’s mental health and to identify the factors asociated with depressive symptoms, anxiety and fear of delivery, which might be something of real interest, but the study should have been conducted on more pregnant women and on a larger period of time.
The manuscript is original and well defined and the results provide an advance in current knowledge. Also,the results are being interpreted appropriately and are significant, but the writting techniques and the structure of the phrases are not well designed, and therefore, it is very difficult to read the article and to understand the meaning of the study. So, unfortunatelly the article is not written in an appropiate way.
The data are robust enough, but it is very difficult to follow the conclusions.
Surely the paper will attract a wide readership, but only with the corrections being made.
The English language is not appropriate and has many writting mistakes, and so, it is imperative to be corrected so that the article could be well designed.
I have some things to add in the lines below, but unfortunatelly, from my point of view, the article must be rewritten more carefully, especially regarding the English language:
Line 14: without „,” before „included”
Line 15: control, not „controls”
Line 18: The COVID-19 Pandemic, not „Pandemic”
Line 19: COVID-19, not „COVID 19”
Line 23: fetal, not „foetal”
Line 28: „,” after „infections”
Line 33: were put, not „have been put”
Line 33: to reduce the infections and the spreading of SARS-CoV2, not „to reduce infections and reduce the spread of SARS-CoV2”
Line 36: „,” after „COVID-19”
Line 36: who were given special care, not „and were given special care”
Line 38: might be, not „may be”
Line 42: before COVID-19 Pandemic, not „before COVID”
Line 45: experienced, not „may experience”
Line 45: that giving birth, not „giving birth”
Line 46: with an increased risk, not „with increased risk”
Line 47: at higher risk, not „at greater risk”
Line 51: should be, not „could be”
Line 55: the level of, not „level of”
Line 59: 1664, not „one-thousand-six-hundred-sixty-four”
Line 59: delivering in the, not „delivering the”
Line 63: no contraindications for, not „no contraindication to”
Line 63: and the will to participate, not „and willing to participate”
Line 66: other than, not „other that”
Line 69: „,” after „women”
Line 70: However, not „Hoverer”
Line 70: delivered, not „were delivered”
Line 72: 465, not „four-hundred-sixty-five”
Line 75: consisted in women, not „constituted women”
Line 80: All pregnant women, not „all participants”
Line 80: signed, not „singed”
Line 84: consisted in, not „consists of”
Line 85: at, not „in”
Line 86: Validated tools were, not „A validated tools”
Line 88: „,” before „respondents”
Line 89: and the partner’s support, not „and support from partner”
Line 90: those women delivering, not „those delivering”
Line 91: COVID-19, not „COVID”
Line 96: to evaluate the level of, not „to evaluate level”
Line 97: it was divided, not „it is divided”
Line 98: consisted of, not „consists of”
Line 100: The symptoms included, not „Symptoms include”
Line 101: „;” after „restlessness” and „;” after „mental tension”
Line 104: „,” before „whereas”
Line 113: is a reliable, not „is reliable”
Line 114: consists in, not „consists of”
Line 118: consists in, not „consists of”
Line 119: only 1 space beween „of” and „life”
Line 120: Each question, not „in each question”
Line 126: general, not „genera”
Line 131: includes, not „include”
Line 143: evaluates, not „evaluate”
Line 143: ways, not „way”
Line 145: only 1 space between „.” and „the”
Line 147: support from, not „support for”
Line 165: „,” after „cases”
Line 167: „,” after „anxiety”
Line 168: only 1 space between „.” and „for”
Line 180: pregnant women, not „women pregnant”
Line 187: the number of, not „number of”
Line 231: COVID-19, not „COVID”
Line 257: the psychic, not „psyche”
Line 264: feared of, not „feared”
Line 281: present, not „are present”
Line 301: that might be due to, not „that might the due to”
Line 305: in contrast with the results, not „it contrasts with the results”
Line 313: were more afraid, not „were most afraid”
Line 321: were overcomed, not „were overcome”
Line 365: are, not „is”
Line 367: benefic effects, not „beneficial effects”
Author Response
Reviewer 1.
- Regarding the structure and accuracy of the phrases, unfortunatelly the manuscript lacks in well structured information, and the phrases are not so well designed. Being an, observational retrospective questionary-based analysis, the study followed a group of 465 women, divided in 2 groups, but this may not be sufficient for such a study, as the 2 groups were analysed only in the period of the third wave of COVID-19. The aim of the analysis was to assess the impact of the COVID-19 Pandemic on adult pregnant women and also the post-partum women’s mental health and to identify the factors asociated with depressive symptoms, anxiety and fear of delivery, which might be something of real interest, but the study should have been conducted on more pregnant women and on a larger period of time.
Ad 1. Thank you for your valuable comments. However, we cannot agree that the population is too small. Sample size calculation was performed showing that to detect 10% difference in the prevalence of severe anxiety based on Brief Self-Rating Scale of Depression and Anxiety 441 total subjects are regirded with the power of 90% and 95% Confidence level. Similarly, with the same parameter, the minimum required sample size of 437 was required to detect the difference of 8% in the prevalence of severe peripartum anxiety based on Labor Anxiety Questionnaire scores. That information was added to the Methods section.
- The manuscript is original and well defined and the results provide an advance in current knowledge. Also,the results are being interpreted appropriately and are significant, but the writting techniques and the structure of the phrases are not well designed, and therefore, it is very difficult to read the article and to understand the meaning of the study. So, unfortunatelly the article is not written in an appropiate way.
Ad 2. Thank you for that remark. The MS was sent for external Native English Editing to correct the clearance of the text.
- The data are robust enough, but it is very difficult to follow the conclusions.
Ad 3. Corrected
- Surely the paper will attract a wide readership, but only with the corrections being made.
Ad 4. As suggested, some corrections were made.
- The English language is not appropriate and has many writting mistakes, and so, it is imperative to be corrected so that the article could be well designed.
Ad 5. Corrected as suggested.
- I have some things to add in the lines below, but unfortunatelly, from my point of view, the article must be rewritten more carefully, especially regarding the English language:
Ad 6. Corrected as suggested.
Line 14: without „,” before „included”
Line 15: control, not „controls”
Line 18: The COVID-19 Pandemic, not „Pandemic”
Line 19: COVID-19, not „COVID 19”
Line 23: fetal, not „foetal”
Line 28: „,” after „infections”
Line 33: were put, not „have been put”
Line 33: to reduce the infections and the spreading of SARS-CoV2, not „to reduce infections and reduce the spread of SARS-CoV2”
Line 36: „,” after „COVID-19”
Line 36: who were given special care, not „and were given special care”
Line 38: might be, not „may be”
Line 42: before COVID-19 Pandemic, not „before COVID”
Line 45: experienced, not „may experience”
Line 45: that giving birth, not „giving birth”
Line 46: with an increased risk, not „with increased risk”
Line 47: at higher risk, not „at greater risk”
Line 51: should be, not „could be”
Line 55: the level of, not „level of”
Line 59: 1664, not „one-thousand-six-hundred-sixty-four”
Line 59: delivering in the, not „delivering the”
Line 63: no contraindications for, not „no contraindication to”
Line 63: and the will to participate, not „and willing to participate”
Line 66: other than, not „other that”
Line 69: „,” after „women”
Line 70: However, not „Hoverer”
Line 70: delivered, not „were delivered”
Line 72: 465, not „four-hundred-sixty-five”
Line 75: consisted in women, not „constituted women”
Line 80: All pregnant women, not „all participants”
Line 80: signed, not „singed”
Line 84: consisted in, not „consists of”
Line 85: at, not „in”
Line 86: Validated tools were, not „A validated tools”
Line 88: „,” before „respondents”
Line 89: and the partner’s support, not „and support from partner”
Line 90: those women delivering, not „those delivering”
Line 91: COVID-19, not „COVID”
Line 96: to evaluate the level of, not „to evaluate level”
Line 97: it was divided, not „it is divided”
Line 98: consisted of, not „consists of”
Line 100: The symptoms included, not „Symptoms include”
Line 101: „;” after „restlessness” and „;” after „mental tension”
Line 104: „,” before „whereas”
Line 113: is a reliable, not „is reliable”
Line 114: consists in, not „consists of”
Line 118: consists in, not „consists of”
Line 119: only 1 space beween „of” and „life”
Line 120: Each question, not „in each question”
Line 126: general, not „genera”
Line 131: includes, not „include”
Line 143: evaluates, not „evaluate”
Line 143: ways, not „way”
Line 145: only 1 space between „.” and „the”
Line 147: support from, not „support for”
Line 165: „,” after „cases”
Line 167: „,” after „anxiety”
Line 168: only 1 space between „.” and „for”
Line 180: pregnant women, not „women pregnant”
Line 187: the number of, not „number of”
Line 231: COVID-19, not „COVID”
Line 257: the psychic, not „psyche”
Line 264: feared of, not „feared”
Line 281: present, not „are present”
Line 301: that might be due to, not „that might the due to”
Line 305: in contrast with the results, not „it contrasts with the results”
Line 313: were more afraid, not „were most afraid”
Line 321: were overcomed, not „were overcome”
Line 365: are, not „is”
Line 367: benefic effects, not „beneficial effects”
Ad 6. Corrected as suggested.
Reviewer 2 Report
The authors conducted a study of an interesting topic which is “Does the COVID-19 pandemic affect pregnant women's fear of labor? “. The quality of presentation of the manuscript, the interest of the readers and the scientific soundness all of them are high scored. Methodology and results are clearly presented. The manuscript requires revision / improvement and responses to the reviewer's recommendations and comments on the following issues:
-
Line 27-30: (quote) „The World Health Organization (WHO) declared a COVID-19 pandemic on March 27 11, 2020 with 424,822,073 confirmed infections including 5,890,312 deaths (data on February 22, 2022) [1-3]. In Poland, on the other hand, the number of infections was 5,582,217, 29 including 110,157 deaths [4] .''Do the quoted phrases apply to the general population or pregnant population? Please specify.
-
Paragraph „Introduction'' needs more details about adverse mental health sequelae of COVID-19 pandemic in the pregnant population. Paragraph „Introduction'' should be improved. Useful information can be find in publication related to the topic : J. Clin. Med. 2022, 11(8), 2072; https://doi.org/10.3390/jcm11082072
-
Line 65: (quote) „37+0 gestation weeks”. Please clarify what is meaning „37+0” or the quoted phrase should be rephrased.
-
Line 78-79: (quote) „A paper-pencil questionnaires were handed in to participants twice: up to 7 days 78 before the delivery and one day postpartum.” It is necessary to insert name of „ a paper-pencil questionnaires” used.
-
The "discussion" should be improved. This requires a greater focus on discussing
the results of the study in correspondence with previous publications of other authors
related to the research topic than on the review of scientific publications itself. - It is recommended to create a paragraph "Strength and Limitation".
-
The "Conclusion" paragraph in current version is poor and needs to be more focused
on the study outcomes. The "Conclusion" paragraph should be revised.
Author Response
Reviewer 2
- The authors conducted a study of an interesting topic which is “Does the COVID-19 pandemic affect pregnant women's fear of labor? “. The quality of presentation of the manuscript, the interest of the readers and the scientific soundness all of them are high scored. Methodology and results are clearly presented.
Ad 1. Thank you for your valuable comments. The responses are written below.
- The manuscript requires revision / improvement and responses to the reviewer's recommendations and comments on the following issues:
- Line 27-30: (quote) „The World Health Organization (WHO) declared a COVID-19 pandemic on March 27 11, 2020 with 424,822,073 confirmed infections including 5,890,312 deaths (data on February 22, 2022) [1-3]. In Poland, on the other hand, the number of infections was 5,582,217, 29 including 110,157 deaths [4] .''Do the quoted phrases apply to the general population or pregnant population? Please specify.
Ad 2. That appled to generla poppulation. We corrected the sentence adding “ in the general population”
- Paragraph „Introduction'' needs more details about adverse mental health sequelae of COVID-19 pandemic in the pregnant population. Paragraph „Introduction'' should be improved. Useful information can be find in publication related to the topic : Clin. Med.2022, 11(8), 2072
Ad 3. Corrected
4. Line 65: (quote) „37+0 gestation weeks”. Please clarify what is meaning „37+0” or the quoted phrase should be rephrase
Ad 4. That mees that it was a term delivery (ended 37 weens). That how it is defined in obstetrics. However, to make it more clear we added (“delivering at term”).
5. Line 78-79: (quote) „A paper-pencil questionnaires were handed in to participants twice: up to 7 days 78 before the delivery and one day postpartum.” It is necessary to insert name of „ a paper-pencil questionnaires” used.
Ad 5. This was a study questionary containing add scales describe below not a specific questionnaire (meaning scale like HADS). That was cleared out in the text
6. The "discussion" should be improved. This requires a greater focus on discussing
the results of the study in correspondence with previous publications of other authors
related to the research topic than on the review of scientific publications itself.
Ad 6. Corrected as requested.
7. It is recommended to create a paragraph "Strength and Limitation".
Ad 7. Added
8. The "Conclusion" paragraph in current version is poor and needs to be more focused
on the study outcomes. The "Conclusion" paragraph should be revised.
Ad 8. Corrected.
Reviewer 3 Report
The study seems ill-equipped to answer the intended research questions.
Specific comments:
1. "The risk of mental illness is significantly increased during pregnancy" - There are important theoretical backgrounds missing in the introduction. It has often been thought that pregnancy is protective against the development of depression, primarily because of the lower suicide rate during pregnancy and during the 2 years after giving birth (citation: pubmed.ncbi.nlm.nih.gov/14519602). In contrast, the postpartum time period clearly was a period of increased risk for the development of MDD (citation: pubmed.ncbi.nlm.nih.gov/22860768). Moreover, a recent study found pregnancy to be associated with a reduced risk for depressive symptoms during the pandemic (citation: ncbi.nlm.nih.gov/pmc/articles/PMC8072624). This was attributed to increased partner support, healthy behaviors, and positive appraisal of the pregnancy.
2. Please change "questionary-based cohort study" to "questionnaire-based cohort study".
3. "... retrospective, questionary-based cohort study" - the study design requires further clarification. Why was it retrospective in nature? Were the questionnaires already routinely put in place for patients prior to the study? Or was this part of a broader quality improvement effort? Please explain.
4. "The questioner consists" - spelling error.
5. Were translated copies of the questionnaire used? Please specify.
6. "465 female participants completed" - do not start a sentence with a number.
7. Please standardize the use of the abbreviation "COVID-19" throughout the manuscript. Some of them are "COVID" and some are "Covid 19".
8. "... wave 3 of the pandemic" - some comments on the local COVID-19 situation on the ground during the period the survey was conducted would be helpful as well. If the study was implemented during peak COVID periods, the findings may have been an anomaly (but may not be of course). It may be useful to plot the trend in cases over time as well so readers have a better understanding of the context.
9. "Hoverer, 83 and 97" - spelling error.
10. Under the discussion of study implications, as resources could be particularly scarce during a serious pandemic situation, timely psychological support could also take many forms, including telemedicine and informal support groups (citation: pubmed.ncbi.nlm.nih.gov/32380875). This should be mentioned.
11. Discussion of study limitations was incomplete.
12. Did the authors adjust for baseline depression or anxiety as a covariate? Additionally, socioeconomic status still varies over time in this age range.
Author Response
Reviewer 3.
- The study seems ill-equipped to answer the intended research questions.
Ad 1. Thank you for your valuable remarks. All answers are written below.
Specific comments:
2. "The risk of mental illness is significantly increased during pregnancy" - There are important theoretical backgrounds missing in the introduction. It has often been thought that pregnancy is protective against the development of depression, primarily because of the lower suicide rate during pregnancy and during the 2 years after giving birth (citation: pubmed.ncbi.nlm.nih.gov/14519602). In contrast, the postpartum time period clearly was a period of increased risk for the development of MDD (citation: pubmed.ncbi.nlm.nih.gov/22860768). Moreover, a recent study found pregnancy to be associated with a reduced risk for depressive symptoms during the pandemic (citation: ncbi.nlm.nih.gov/pmc/articles/PMC8072624). This was attributed to increased partner support, healthy behaviors, and positive appraisal of the pregnancy.
Ad 2. We cannot agree with these arguments. All national and international obstetrics and gynecological association recommend screening for depression symptoms in pregnancy (please seeMikšić Š, Miškulin M, Juranić B, Rakošec Ž, Včev A, Degmečić D. Depression and Suicidality during Pregnancy. Psychiatr Danub. 2018 Mar;30(1):85-90. doi: 10.24869/psyd.2018.85. PMID: 29546863. as an example). Definitely pregnancy is not a protective factors according to EMB (please see for example Cleveland Clinic Journal of Medicine May 2020, 87 (5) 273-277;DOI: https://doi.org/10.3949/ccjm.87a.19054 and Learman LA. Screening for Depression in Pregnancy and the Postpartum Period. Clin Obstet Gynecol. 2018 Sep;61(3):525-532. doi: 10.1097/GRF.0000000000000359. PMID: 29389681). As the etiology of depression in pregnancy has been well described and the subject of this study is not a depression but anxiety, we decided not to add any literature on this subject.
We also read the articled that was cited by the reviewer (ncbi.nlm.nih.gov/pmc/articles/PMC8072624)., However there are some methodological biases in this text. Firstly, scores in PHQ2 did not differ significantly between pregnant and not-pregnant women. Secondly the model with pregnancy as protective factor explained only 9% of the variance in depressive symptoms. And finally, the authors used raw scores for scale not cut-off score; so it can be said that pregnancy may reduce the scores in that scale but from this analysis we do not know if it influence the prevalence of clinically significant depressive symptoms. For all that reasons we decided not to discuss that in the introduction section.
3. Please change "questionary-based cohort study" to "questionnaire-based cohort study".
Ad 3. Changed
4. "... retrospective, questionary-based cohort study" - the study design requires further clarification. Why was it retrospective in nature? Were the questionnaires already routinely put in place for patients prior to the study? Or was this part of a broader quality improvement effort? Please explain.
Ad 4. Thank you for that remark. It is not a repressive study as we evaluated the anxiety just before the the delivery. We though have not assessed the anxiety retrospectively after the delivery. That uncleanse was cleared out in the text by deleting “retrospective”
5. "The questioner consists" - spelling error.
Ad 5. Corrected.
6. Were translated copies of the questionnaire used? Please specify.
Ad 6. All scales used in the questionnaire were validated in Poland.
7. "465 female participants completed" - do not start a sentence with a number.
Ad 7. Corrected.
8. Please standardize the use of the abbreviation "COVID-19" throughout the manuscript. Some of them are "COVID" and some are "Covid 19".
Ad 8. Corrected
9. "... wave 3 of the pandemic" - some comments on the local COVID-19 situation on the ground during the period the survey was conducted would be helpful as well. If the study was implemented during peak COVID periods, the findings may have been an anomaly (but may not be of course). It may be useful to plot the trend in cases over time as well so readers have a better understanding of the context.
Ad 9. That was slightly changed in the text – “However, as the number of COVID-19 positive cases in general population and in pregnant women did not differ significantly between 3rd and 2nd waves, that have not biased the results”.
10. "Hoverer, 83 and 97" - spelling error.
Ad 10. corrected
11. Under the discussion of study implications, as resources could be particularly scarce during a serious pandemic situation, timely psychological support could also take many forms, including telemedicine and informal support groups (citation: pubmed.ncbi.nlm.nih.gov/32380875). This should be mentioned.
Ad 11. Added
12. Discussion of study limitations was incomplete.
Ad 12. Corrected
13. Did the authors adjust for baseline depression or anxiety as a covariate? Additionally, socioeconomic status still varies over time in this age range.
Ad 13. Yes, trait anxiety and baseline depression were used as covariates as well as socioeconomical status.
Round 2
Author Response
Thank you for reviewing the following research paper.
Reviewer 2 Report
The authors referred to some but not to all the reviewer's comments and did not improve the manuscript to a satisfactory level. In example, the "Conclusion" paragraph is not revised and thus it is poor still and needs to be more focused on the study outcomes. The "Conclusion" paragraph should be revised / improved.
Author Response
Thank you for reviewing the following research paper. According to the recommendations, the conclusions have been changed to follow from the results.

Reviewer 3 Report
The manuscript is still in need of extensive edits for language and style.
Specific comments:
1. The authors seemed to have misunderstood my earlier point. My point was that pregnancy per se is not associated with increased risk of mental disorders, rather there are specific predisposing factors for poorer mental health, e.g. young, unmarried women with recent stressful life events, complicated pregnancies, and poor overall health. These groups have been shown to have a significantly increased risk of mental disorders during pregnancy (citation: ncbi.nlm.nih.gov/pmc/articles/PMC2669282).
2. "All scales used in the questionnaire were validated in Poland" - so were Polish versions of the questionnaires used? Please specify.
3. Please change "all of this has not" to "all of these have not".
Author Response
Ad. 1
Thank you for your valuable comment.
I agree that the mentioned risk factors increase the chance of developing depression during pregnancy. Among them are: anxiety, low social support, domestic violence, etc (https://www.ncbi.nlm.nih.gov/pmc/articles/PMC2919747/).
I read the quoted article mentioned that pregnancy in itself does not increase the risk of depression. However, in another paper, the authors state in their conclusions that pregnancy does not protect against the development of depression (https://pubmed.ncbi.nlm.nih.gov/16121830/).
The most important element is the elimination of risk factors, proper screening diagnosis and treatment during pregnancy.
Ad.2
Yes, scales that have been validated for the Polish language have been used. Respectively:
- Brief Self-Rating Scale of Depression and Anxiety (BSRSDA)- Krótka Skala Samooceny Depresji i Lęku Andrzej Kokoszka- https://www.termedia.pl/Krotka-Skala-Samooceny-Depresji-i-Leku-opis-konstrukcji-oraz-wlasciwosci-psychometryczne-dla-osob-z-cukrzyca,8,11912,0,0.html
- Labor Anxiety Questionnaire (KLP II)- Kwestionariusz Lęku Porodowego (KLP II) Wersja Zrewidowana. L. Putyński, M. Paciorek KLP II
- Rosenberg Self-Esteem Scale (SES)- Skala Samooceny SES M. Rosenberga w polskiej adaptacji I. Dzwonkowskiej, K. Lachowicz-Tabaczek i M. Łaguny
- Communication in Marriage Questionnaire (CMQ)- Kwersionariusz komunikacji małżeńskiej -KKM (ocena własnych zachowań ); (ocena zachowań partnera ) Maria Kaźmierczak, Mieczysław Plopa
- The State-Trait Anxiety Inventory (STAI)- Kwestionariusz Samooceny STAI, ARKUSZ X-1, ARKUSZ X-2; C.D. Spielberger, J. Strelau, M. Tysarczyk, K. Wrześniewski
- Berlin Social Support Scale (BSSS) - Berlińska Skala Wsparcia Społecznego (BSSS) Aleksandra Łuszczyńska, Monika Kowalska , Ralf Schwarzer & Ute Schulz, 2002; Freie Universität Berlin, Health Psychology
- Provisions of Social Relation Scale (PSRS), from partner – Spostrzegane wsparcie społeczne (oparta na skali Provisons of Social Relation Scale; Turner, Marino, 1994) (adaptacja polska: Kaniasty, 2002)
- The Satisfaction with Life Scale (SWLS)- Skala Satysfakcji z życia- E. Diener, R.A. Emmons, R.J. Larson I S. Griffin
- Hospital Anxiety Depression Scale (HADS)- Szpitalnej Skali Leku i Depresji; Zigmond AS, Snaith RP. The hospital anxiety and depression scale. Acta Psychiatrica Scandinavica 1983; (67): 361 –370.; de Walden –Gałuszko K, Majkowicz M. Ocena jakości opieki paliatywnej w teorii i praktyce. Wyd. Akademia Medyczna, Zakład Opieki Paliatywnej, Gdańsk 2000.]
Ad.3
Corrected
MS was previously revised by an outside Native English Editing.
